# Predictions of the Wettable Parameters of an Axisymmetric Large-Volume Droplet on a Microstructured Surface in Gravity

**DOI:** 10.3390/mi14020484

**Published:** 2023-02-19

**Authors:** Jian Dong, Jianliang Hu, Zihao Zhang, Mengying Gong, Zhixin Li

**Affiliations:** 1Key Laboratory of E&M, Zhejiang University of Technology, Hangzhou 310023, China; 2State Key Lab of Transducer Technology, Shanghai Institute of Microsystem and Information Technology, Chinese Academy of Sciences, Shanghai 200031, China

**Keywords:** wettable parameters, axisymmetric large-volume droplet, gravity, microstructured horizontal plane, external spherical surface

## Abstract

In this study, a numerical model was developed to predict the wettable parameters of an axisymmetric large-volume droplet on a microstructured surface in gravity. We defined a droplet with the Bond number $B_{o}>0.1$ as a large-volume droplet. $B_{o}$ was calculated by using the equation Bo=ρlgγlv3V4π23 where $\rho_{l}$ is the density of liquid, $\gamma_{\mathrm{lv}}$ is the liquid-vapor interfacial tension, g is the gravity acceleration and V is the droplet volume. The volume of a large-volume water droplet was larger than 2.7 μL. By using the total energy minimization and the arc differential method of the Bashforth–Adams equation, we got the profile, the apparent contact angle and the contact circle diameter of an axisymmetric large-volume droplet in gravity on a microstructured horizontal plane and the external spherical surface. The predictions of our model have a less than 3% error rate when compared to experiments. Our model is much more accurate than previous ellipsoidal models. In addition, our model calculates much more quickly than previous models because of the use of the arc differential method of the Bashforth–Adams equation. It shows promise for use in the design and fabrication of microfluidic devices.

## 1. Introduction

The study of an axisymmetric large-volume droplet on a microstructured surface in gravity (an axisymmetric heavy droplet) is a fundamental problem in the mechanics of wetting and spreading that facilitates a better understanding of how to modify the wettable property of the rough surface. It attracted a lot of research due to its applications in microfluidic devices, microelectromechanical systems (MEMS), medical devices and instrumentation, chemical engineering and bioengineering [1,2,3,4,5,6]. The wettable parameters of an axisymmetric heavy droplet depend on the following influence factors: (1) the mass of the droplet; (2) the topology structure and roughness of the surface; (3) the interfacial tensions among the droplet liquid, the vapor and the surface material. Other works studying the wettable parameters of an axisymmetric heavy droplet were done. Tammar S. Meiron [7] used an optical system to get the vibrating droplets’ profiles and calculated the apparent contact angles (ACAs) of axisymmetric heavy droplets. Chong Li [8] used phase-field interface tracking to simulate the wetting of droplets on rough surfaces and calculated the ACAs of axisymmetric heavy droplets. Regarding the periodic distribution of regular microstructures on surfaces (regular rough surfaces), many researchers performed studies to simplify the calculation of wettable parameters of axisymmetric heavy droplets. H. Yildirim Erbil [9], Edward Bormashenko [10], Vlado A. Lubarda [11] and W. Xue Wei [12] used ellipsoidal models to gain the approximate shapes and ACAs of an axisymmetric heavy droplet on smooth surfaces from 2000 to 2015. However, the ellipsoidal models keep in agreement with practical situations only when $ACA<120^{\circ }$. M. Hajirahimi [13] used the direct integration of the Laplace equation to gain the profiles and ACAs of an axisymmetric heavy droplet on smooth spherical surfaces in 2015. Based on Matlab’s “fmin” function, Jian Dong [14] used the direct discretion of the droplet profile to gain the profiles and ACAs of an axisymmetric heavy droplet on rough horizontal surfaces in 2017. All of those models mentioned above are accurate in their special conditions, but they are not universal.

In this paper, we put forward a more universal, accurate and quick-to-calculate numerical model to calculate the wettable parameters of an axisymmetric heavy droplet on regular rough surfaces in all conditions, no matter whether the surface is horizontal or special, whether the surface is smooth or rough, and whether its ACA is less than 120°. We did the following: (1) Based on the Bashforth–Adams equation and the arc differential method, we gained the admissible profile cluster of an axisymmetric heavy droplet; (2) based on the total energy minimization, we found out the real profile of an axisymmetric heavy droplet and calculated the wettable parameters; (3) we did experiments to verify the results from our model and found that the model results are in good agreement with the experimental results; (4) we found that our model is more accurate, universal and quick-to-calculate than previous ellipsoidal models. The work presented in this paper provides a better method for calculating the wetting parameters of an axisymmetric heavy droplet, which is expected to be used in the design and fabrication of microfluidic devices.

## 2. Theoretical Models

### 2.1. Bashforth–Adams Equation and Its Arc Differential Solution

As shown in Figure 1, O is the coordinate origin, x and z are the horizontal and vertical coordinates of a point S on the droplet external surface and ϕ is the tangential angle of the point S.

Bashforth and Adams derived the Bashforth–Adams equation (B–A equation) based on the Young–Laplace [15,16] formula, which describes an equation for the axisymmetric heavy droplet at static equilibrium, as is shown in Equation (1):(1)$$\frac{z^{\prime \prime }}{{\left(1+z^{\prime }{}^{2}\right)}^{\frac{3}{2}}}+\frac{z^{\prime }}{x{\left(1+z^{\prime }{}^{2}\right)}^{\frac{1}{2}}}=\frac{2}{b}+\frac{(\rho_{1}-\rho_{v})gz}{\gamma_{\mathrm{lv}}}$$
where $\rho_{1}$ and $\rho_{v}$ are the density of liquid and air, respectively, $\gamma_{\mathrm{lv}}$ is the liquid-vapor interfacial tension, g is the gravity acceleration and b is the curvature radius at the droplet navel. b is named as the size factor, and $\beta =\frac{\left(\rho_{l}-\rho_{v}\right)gb^{2}}{\gamma_{\mathrm{lv}}}$ is named as the shape factor.

Neumann et al. [17,18,19] rectified the B–A equation into a first-order arc differential equation (ARC–B–A equation) as follows:(2)$$\frac{d\phi }{ds}+\frac{\sin\phi }{x}=\frac{2}{b}+\frac{(\rho_{1}-\rho_{v})gz}{\gamma_{\mathrm{lv}}}$$
where s is the arc length measured from the origin O and ds is the arc differential.

As shown in Figure 1, a geometrical consideration yields the following differential identities:(3)$$\frac{dx}{ds}=\cos\phi \;\frac{dz}{ds}=\sin\phi$$

In the X–O–Z plane, a suitable representation of the droplet profile is the parametric form below:(4)x=x(s), z=z(s), ϕ=ϕ(s)

In this representation, both x and z are single-valued functions of s. The droplet profile has the boundary conditions
(5)x(0)=z(0)=ϕ(0)=0

For the given values $\rho_{l},\rho_{v},\gamma_{\mathrm{lv}}$ and b(β), the droplet profile can be obtained by integrating ds simultaneously. Figure 2 shows the profiles of droplets in the ϕ range of 0–180° with different b(β) values. The profiles are a family of bundle center curves, which forms the admissible profile space of the axisymmetric heavy droplets.

### 2.2. The Total Energy of an Axisymmetric Heavy Droplet on the Micro-Nano Structured Surface

The total energy of an axisymmetric heavy droplet on the micro-nano structured surface includes the external surface energy, the internal surface energy and the gravity potential energy. The total energy [20,21,22] in the Wenzel or Cassie state, $E_{\mathrm{Wenzel}}$ or $E_{\mathrm{Cassie}}$, can be respectively expressed as follows ($\rho_{v}$ can be neglected when comparing with $\rho_{l}$):

Wenzel state:(6)$$E_{\mathrm{Wenzel}}=\gamma_{\mathrm{lv}}\iint_{S_{\mathrm{ext-W}}}dS+\left(\gamma_{\mathrm{lv}}-\gamma_{\mathrm{sv}}\right)r_{\mathrm{gh}}\iint_{S_{\mathrm{base-W}}}dS+\underset{\Omega_{W}}{∭}\rho_{l}gzdV$$

Cassie state:(7)$$E_{\mathrm{Cassie}}=\gamma_{\mathrm{lv}}\iint_{S_{\mathrm{ext-C}}}dS+\left[f\gamma_{\mathrm{sl}}+\left(1-f\right)\gamma_{\mathrm{lv}}-f\gamma_{\mathrm{sv}}\right]r_{\mathrm{gh}}\iint_{S_{\mathrm{base-C}}}dS+\underset{\Omega_{C}}{∭}\rho_{l}gzdV$$
where γ is the interfacial tension that gives lv,sl and sv, which represent liquid-vapor, solid-liquid and solid-vapor, respectively, $r_{\mathrm{gh}}$ is the roughness factor of the base, f is the area fraction of the base, $\rho_{l}$ is the density of liquid, g is the gravity acceleration, $\Omega_{C}$ is the Cassie droplet body and $\Omega_{W}$ is the Wenzel droplet body. $S_{\mathrm{ext-W}},S_{\mathrm{base-W}},S_{\mathrm{ext-C}}$ and $S_{\mathrm{base-C}}$ represent the external and internal (ext- and base-, respectively) surfaces of the Wenzel and Cassie droplets (W and C, respectively).

### 2.3. The Real State and the Possible States of an Axisymmetric Heavy Droplet with a Fixed Volume

An axisymmetric heavy droplet with a fixed volume has many possible states, each of which occupy their own total energies. However, the real state occupies the minimum total energy. For example, as shown in Figure 3, E1, E2 and E3 are three possible states of the droplet. However, E2 is the minimum total energy state in all possible states. Thus, state 2 is the real state of the droplet.

## 3. Numerical Methods

### 3.1. Algorithm

Figure 2 gives the admissible profile space with the parameter b of the axisymmetric heavy droplets’ profiles. For the given values of $\rho_{1},\gamma_{\mathrm{lv}}$ and the droplet volume V, the sealed profiles of the axisymmetric heavy droplets are only determined by b. If we find out the minimum total energy with volume constraint in all possible droplets’ states with different values of b, then we can find the real sealed profile of the axisymmetric heavy droplet on the microstructured surface.

The curvature radii at droplet navels b are a dense and complete set of real numbers. The ordinary differential equation forms a continuous single shot from b to the heavy droplet profile curvatures. Thus, the space of the heavy droplet profile curvatures is dense and complete. The Lagrange multiplier method can be used to search all possible states of the heavy droplet to solve for the minimum energy.

Equations (2)–(5) demonstrate that the droplet profile question is a Cauchy problem of the ARC–B–A equation, which is a first-order ordinary differential equation related to the arc differential ds. We can use the arc difference Δs method to numerically calculate out the coordinates and parameters ($x_{j},z_{j},\phi_{j}\;j=1,2,3...$) of every difference point on the profile. The difference expression of the droplet volume on the horizontal plane $V_{\mathrm{c-p}}$, the difference expression of the droplet volume on the external spherical surface $V_{\mathrm{c-s}}$, the expression of the area of the solid-liquid interface on the horizontal plane $S_{\mathrm{base-p}}$, the expression of the area of the solid-liquid interface on the horizontal plane $S_{\mathrm{base-s}}$, the total energy of a Wenzel droplet on a horizontal plane $E_{\mathrm{Wenzel-p}}$, and the external spherical surface, the total energy of a Cassie droplet on a horizontal plane $E_{\mathrm{Cassie-p}}$, the total energy of a Wenzel droplet on an external spherical surface $E_{\mathrm{Wenzel-s}}$ and the total energy of a Cassie droplet on an external spherical surface $E_{\mathrm{Cassie-s}}$ can be expressed as follows:(8)$$S_{\mathrm{base-p}}=\pi x_{v}^{2}$$
(9)$$S_{\mathrm{base-s}}=2\pi R(R-\sqrt{R^{2}-x_{v}^{2}})$$
(10)$$V_{\mathrm{c-p}}=\sum_{j=1}^{N}\pi x_{j}^{2}\left(z_{j}-z_{j\text{-}1}\right)$$
(11)$$V_{\mathrm{c-s}}=\sum_{j=1}^{N}\pi x_{j}^{2}\left(z_{j}-z_{j\text{-}1}\right)-\frac{\pi \left(3R{\left(R-\sqrt{R^{2}-x_{j}^{2}}\right)}^{2}-{\left(R-\sqrt{R^{2}-x_{j}^{2}}\right)}^{3}\right)}{3}$$
(12)$$E_{\mathrm{Wenzel-p}}=2\pi \gamma_{\mathrm{lv}}\Delta s\sum_{j=1}^{N_{v}}x_{j}+\pi r_{\mathrm{gh}}\left(\gamma_{\mathrm{lv}}-\gamma_{\mathrm{sv}}\right)x_{v}^{2}+\pi \rho_{l}g\sum_{j=1}^{N_{v}}\left(z_{v}-z_{j}\right)x_{j}^{2}\left(z_{j}-z_{j\text{-}1}\right)$$
(13)$$E_{\mathrm{Cassie-p}}=2\pi \gamma_{\mathrm{lv}}\Delta s\sum_{j=1}^{N_{v}}x_{j}+\pi \left[f\gamma_{\mathrm{sl}}+\left(1-f\right)\gamma_{\mathrm{lv}}-f\gamma_{\mathrm{sv}}\right]x_{v}^{2}+\pi \rho_{l}g\sum_{j=1}^{N_{v}}\left(z_{v}-z_{j}\right)x_{j}^{2}\left(z_{j}-z_{j\text{-}1}\right)$$
(14)$$E_{\mathrm{Wenzel-s}}=2\pi \gamma_{\mathrm{lv}}\Delta s\sum_{j=1}^{N_{v}}x_{j}+2\pi r_{\mathrm{gh}}R(R-\sqrt{R^{2}-x_{v}^{2}})\left(\gamma_{\mathrm{lv}}-\gamma_{\mathrm{sv}}\right)+\pi \rho_{l}g\sum_{j=1}^{N_{v}}\left(z_{v}-z_{j}\right)x_{j}^{2}\left(z_{j}-z_{j\text{-}1}\right)$$
(15)$$E_{\mathrm{Cassie-s}}=2\pi \gamma_{\mathrm{lv}}\Delta s\sum_{j=1}^{N_{v}}x_{j}+2\pi R(R-\sqrt{R^{2}-x_{v}^{2}})\left[f\gamma_{\mathrm{sl}}+\left(1-f\right)\gamma_{\mathrm{lv}}-f\gamma_{\mathrm{sv}}\right]+\pi \rho_{l}g\sum_{j=1}^{N_{v}}\left(z_{v}-z_{j}\right)x_{j}^{2}\left(z_{j}-z_{j\text{-}1}\right)$$
where R is the radius of the sphere and $z_{v}$ and $x_{v}$ are the height and the contact circle radius of the droplet, respectively, solved at the volume constraint. For a plane decorated with circular microstructures, $r_{\mathrm{gh}}$ and f can be expressed by the periodic spacing a, the diameter d and the height h as follows:(16)$$r_{\mathrm{gh}}=1+\frac{\pi dh}{{\left(a+d\right)}^{2}}$$
(17)$$\;f=\frac{\pi d^{2}}{4{\left(a+d\right)}^{2}}\;$$

### 3.2. The Flow Chart

According to the algorithm mentioned above, the flow chart for the calculation of the wettable parameters of the axisymmetric heavy droplet is shown in Figure 4.

Briefly, the process is as follows: set the initial value of the droplet size factor b as the droplet equivalent radius $b_{0}$; use the arc difference method to get the coordinates of every difference point on the profile; use volume constraint to define the position of the solid-liquid interface ($z_{v}$); calculate out the total energy with $b_{0}$; step forward b by Δb; repeat the aforementioned process; calculate total energy until the calculated total energy begins to increase with b; record the minimum energy state and corresponding ACA, $r_{b}$ and h. *PS* is a symbol of the shape of the solid-liquid contact area and CW is the symbol of the droplet wetting state.

## 4. Experiments

### 4.1. Fabrication of the Si Rough Surface

The fabrication process for the Si rough surface started from 4-inch n-type (100) silicon wafers (Figure 5). First, the AZ4620 photoresist (Qiyao, Shenzhen, China )was patterned for the diameter d and the periodic spacing a of the periodic circular microstructures. Second, deep reactive ion etching (DRIE) was processed for the height h of the circular microstructures. Third, the photoresist was striped in wet solution, and we got Si microstructures on the surface. The deep reactive iron etching (DRIE) process used passivation and etching cycles, with a flow rate of 85 sccm for C4F8 in the passivation cycle and 85 sccm for SF6 in the passivation cycle.

### 4.2. Characterization for Droplets’ Wettable Parameters on Microstructured Surfaces

As shown in Figure 6, we used SDC-80 (Sindin, Dongguan, China) profile and contact angle measurement to characterize the droplets’ wettable parameters on microstructured surfaces. Deionized water and glycerol droplets were emitted from a needle onto the rough surface. A video camera was set on side of the droplet, and the optical axis of the camera was perpendicular to the droplet. After emitting droplets for ten minutes, we took pictures of the equilibrium droplets with different volumes and recorded the profiles, contact circle diameters and ACAs.

## 5. Results and Discussion

### 5.1. Pictures from Our Model Compared with Those from Experiments and the Ellipse Model

As shown in Figure 7, picture numbers 1–12 are respectively related to the 12 wettable types in Table 1. In all conditions, red lines coincided exactly with yellow lines. Blue lines coincided with yellow lines only when $ACA<120^{\circ }$, whereas the blue lines largely deviated from the yellow lines when $ACA\geq 120^{\circ }$. This means our model adapts to all conditions, which means it is more accurate and universal than the ellipse model [11,12,23,24].

The relative structure spacing a/d and the relative structure height h/d are important for the values of the contact angle. This was demonstrated in our previous study [14].

### 5.2. Result Comparations in Different Droplet Bond Numbers

Figure 8 further shows the excellent universality and good accuracy of our model. No matter how smooth or rough, hydrophobic or hydrophilic, horizontally flat or spherical the surfaces were, our model data coincided highly with experimental data with different values of $B_{o}$ from 0.3 to 1.2. Conversely, the ellipse model was not accurate when $ACA\geq 120^{\circ }$, which limits its use on superhydrophobic surfaces.

## 6. Conclusions

By using the total energy minimization and the arc differential method of the Bashforth–Adams equation, we set up theoretical models and related numerical methods to obtain the wettable parameters of axisymmetric large-volume droplets on microstructured horizontal planes and external spherical surfaces. The profiles, contact circle diameters and ACAs were calculated. Experimental results were used to verify the correctness of our model, which shows our model coincided highly with the experiments. The maximum error of our model’s results to those of the experiments did not exceed 3%. In addition, our model can be calculated at high speed and is widely applicable. This study provides a good theoretical basis and an excellent quantitative method for the design and fabrication of functional surfaces in microfluidic chips.

## Figures and Tables

**Figure 1 micromachines-14-00484-f001:**
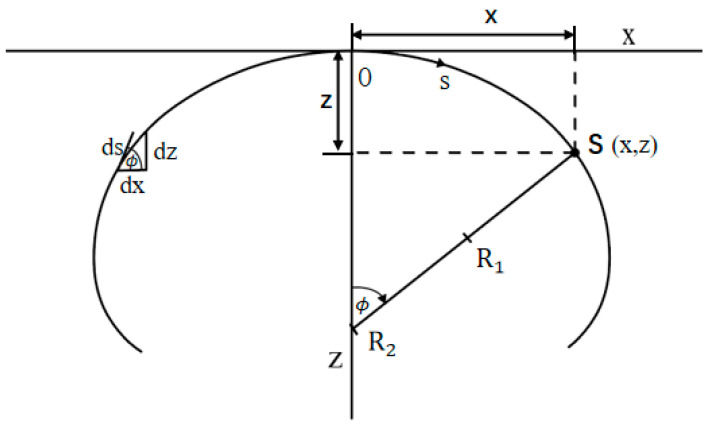
Schematic diagram of an axisymmetric heavy droplet and its geometrical parameters.

**Figure 2 micromachines-14-00484-f002:**
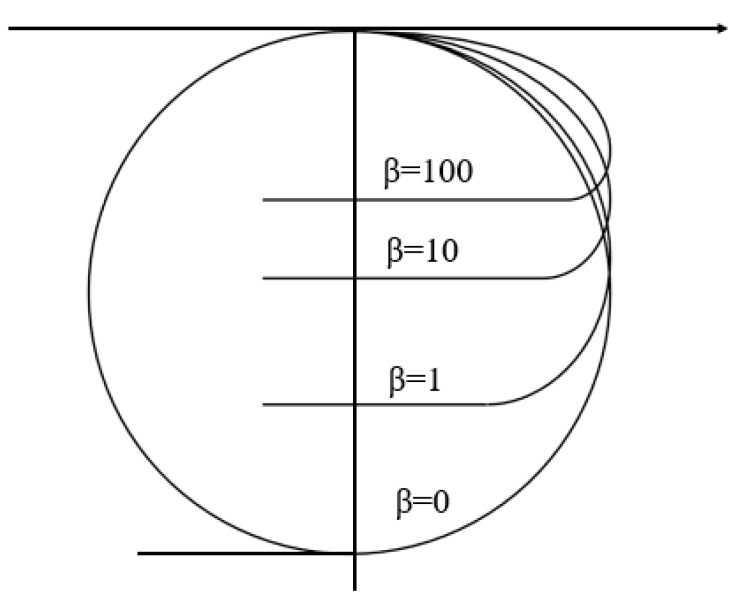
A family of axisymmetric heavy droplet profiles with different b(β).

**Figure 3 micromachines-14-00484-f003:**
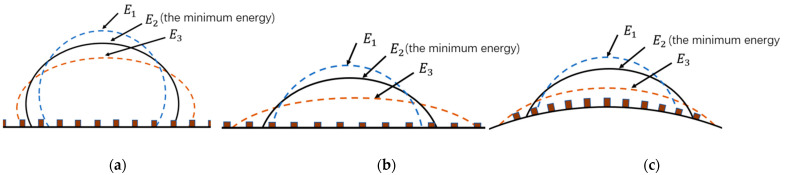
E2 is the minimum total energy in all possible total energies; thus, state 2 is the real state. (**a**) An axisymmetric heavy droplet on a microstructured superhydrophobic plane. (**b**) An axisymmetric heavy droplet on a microstructured superhydrophilic plane. (**c**) An axisymmetric heavy droplet on a microstructured superhydrophilic external spherical surface.

**Figure 4 micromachines-14-00484-f004:**
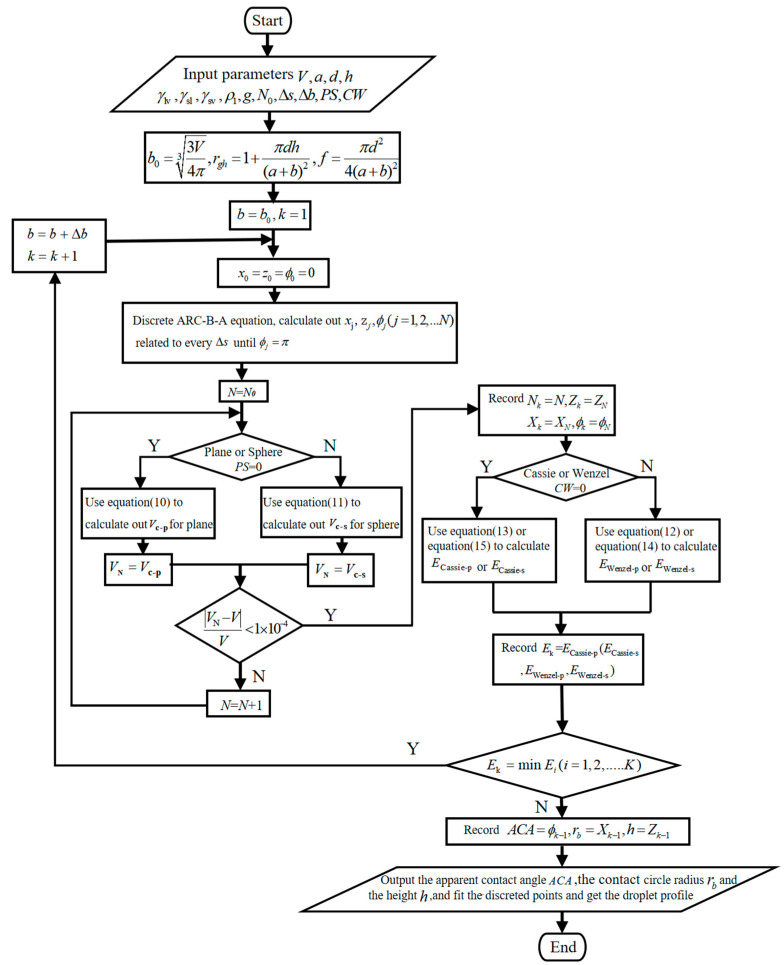
Flow chart of the calculation of the wettable parameters of the axisymmetric heavy droplet.

**Figure 5 micromachines-14-00484-f005:**
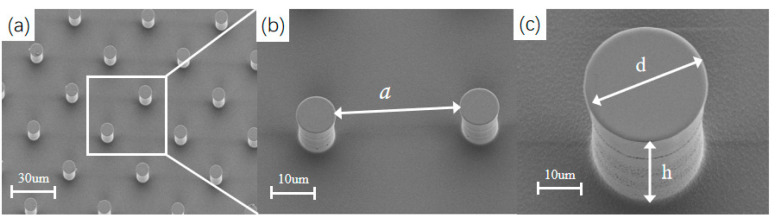
The SEM pictures of Si rough surfaces. (**a**) the whole picture of the Si rough surface. (**b**) the periodic spacing a of the circular microstructures. (**c**) the diameter d and the height h of the circular microstructures.

**Figure 6 micromachines-14-00484-f006:**
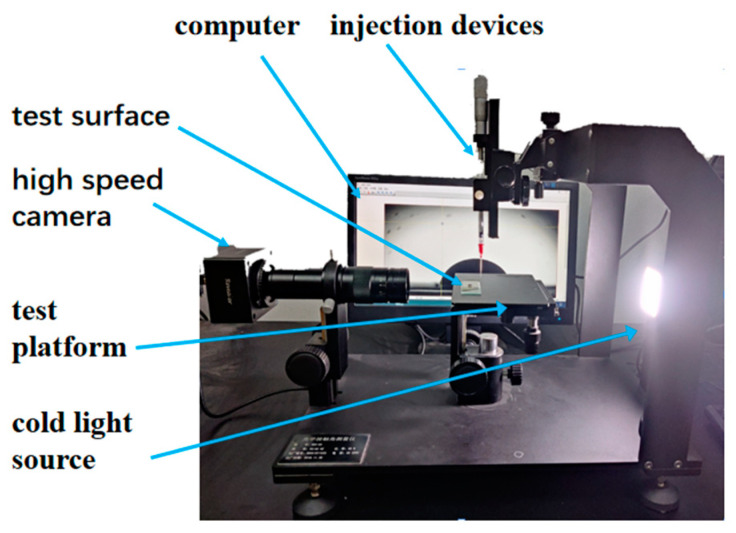
Schematic of the experimental apparatus.

**Figure 7 micromachines-14-00484-f007:**
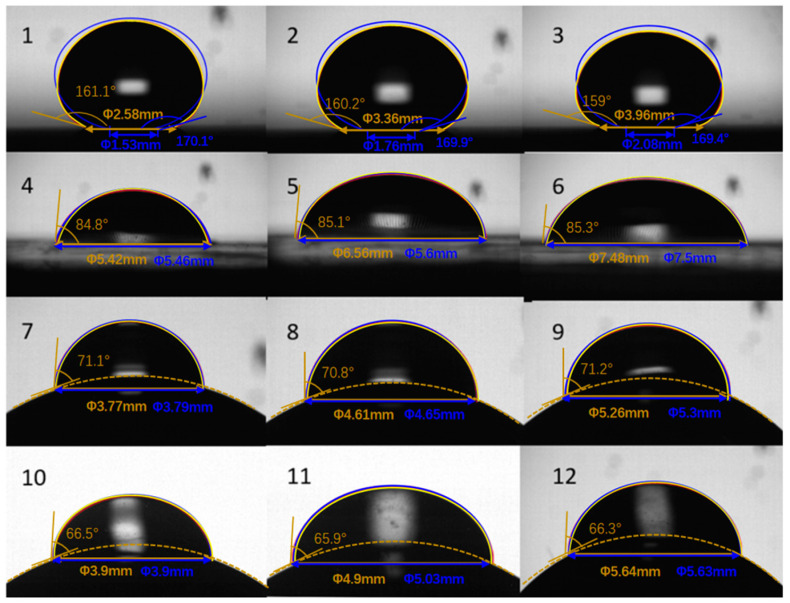
Picture comparations of results from our model, experiments and the ellipse model. Picture numbers 1–12 are respectively related to the 12 wettable types in Table 1. Profiles from our model are shown with red lines, those of experiments with yellow lines and those of the ellipse model with blue lines. The ellipse model is from Lubarda’s paper [11].

**Figure 8 micromachines-14-00484-f008:**
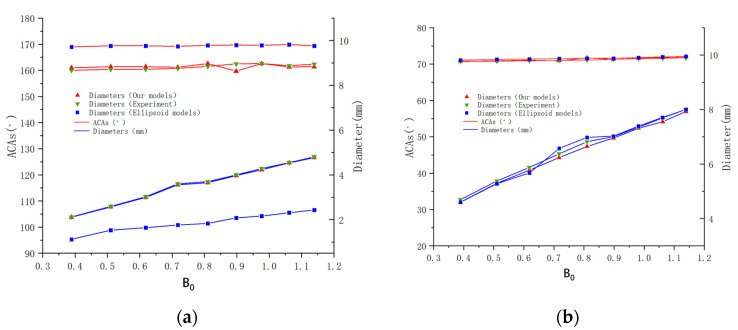
Comparations among results from our model, from experiments and from the ellipse model with different droplet Bond numbers. (**a**) The horizontal microstructured plane (Types 1, 2 and 3) and (**b**) the smooth external spherical surface (Type 7, 8 and 9).

**Table 1 micromachines-14-00484-t001:** 12 Wettable types and results from the experiments, our model and the ellipse model.

Material	Microstructure	Type	Volume (μL)	ACA from Experiments θ (◦)	ACA from Our Model θ (◦)	ACA from tde Ellipse Model θ (◦)	Diameter from Experiments d (mm)	Diameter from Our Model d (mm)	Diameter from tde Ellipse Model d (mm)
Water + Si	d = 10 μm	1	30	160.3	161.1	170.1	2.58	2.592	1.53
	a = 30 μm	2	50	159.1	160.2	169.9	3.36	3.364	1.76
	h = 20 μm	3	70	161.6	159.0	169.4	3.96	3.998	2.08
Glycerin + PDMS	d = 10 μm	4	30	85.1	84.8	83.1	5.42	5.432	5.441
	a = 60 μm	5	50	85.6	85.1	83.8	6.56	6.582	6.600
	h = 20 μm	6	70	86.1	85.3	84.3	7.48	7.490	7.503
Water + iron	R = 6 mm	7	10	72.5	71.1	73.0	3.68	3.770	3.830
		8	20	71.8	70.8	71.6	4.60	4.610	4.630
		9	30	72.5	71.2	73.0	5.26	5.260	5.280
Glycerin + iron	R = 6 mm	10	10	67.5	66.5	68.4	3.90	3.894	3.906
		11	20	66.8	65.9	67.9	4.90	4.904	4.910
		12	30	67.1	66.3	67.3	5.64	5.634	5.640

The surface tension of water is 72.8 mJ/m^2^; Si, 30 mJ/m^2^; glycerin, 63.3 mJ/m^2^; PDMS, 19 mJ/m^2^ and iron, 46 mJ/m^2^. The ellipse model is from Lubarda’s paper [11].

## Data Availability

Not applicable.

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
