# Peer review of "Predictions of the Wettable Parameters of an Axisymmetric Large-Volume Droplet on a Microstructured Surface in Gravity"

_micromachines, 2023, doi:10.3390/mi14020484_

Round 1

Reviewer 1 Report

A numerical model is developed to predict the wettable parameters of an axisymmetric large-volume droplet on a microstructured surface in gravity. By the total energy minimization and the arc differential method of the Bashforth-Adams equation, we get the profile, the apparent contact angle and the contact circle diameter of an axisymmetric large-volume droplet in gravity on the microstructured horizontal plane and the external spherical surface. The predictions are in good agreement with the results from experiments. as such this work is interesting, it can be acceppted for publication

Author Response

Dear Reviewer

Thank you for raising these issues and we have revised the document for your review in the light of your comments. I look forward to hearing from you.

Best regards

Reviewer 2 Report

Dear Authors,

1. The paper is scientifically interesting. I suggest making the following changes to make it better.

2. Lines 10 / 22 - Please specify what is the "large-volume" scale. Enter a value or range of values numerically.

3. Lines 13-14 - Please enter the percentage effectiveness of the model of prediction. The term "good agreement" is too general and relative.

4. The introduction section is insufficient. The bibliographic list could be more extensive. Please write more examples of testing the contact angle of rough surfaces. For example, the following papers address these issues:

https://doi.org/10.1038/s41598-020-80729-9

https://doi.org/10.1016/j.triboint.2021.107139

https://doi.org/10.1016/j.jcis.2004.02.036

5. Line 53 - When we say the drop is heavy and when it's light? What values define it?

6. Please write what parameters of the surface microstructure are important for the prediction of the contact angle? Examples of descriptions of surface microstructure parameters can be found, for example, in:

https://doi.org/10.3390/cryst11111371

https://doi.org/10.1007/s42242-019-00042-x

https://doi.org/10.1016/j.measurement.2022.112066

7. Chapter 4.1. - Provide the key processing parameters that affect the achievement of the appropriate surface microgeometry.

8. Specify the time of dispensing the drop and measuring the drop after deposition on the surface.

Author Response

Dear Reviewer

Thank you for raising these issues and we have revised the document for your review in the light of your comments. I look forward to hearing from you.

Best regards

                            Response to Reviewer 2 Comments

Point 1: Lines 10 / 22 - Please specify what is the "large-volume" scale. Enter a value or range of values numerically.

Response 1: We add “ We define a …larger than 2.7 ul” in lines 10-14 to specify "large-volume".  

Point 2: Lines 13-14 - Please enter the percentage effectiveness of the model of prediction.The term "good agreement" is too general and relative.

Response 2: We add ” have below 3% errors to” in lines 18 to show the maximum error of our model to experiments. 

Point 3: The introduction section is insufficient. The bibliographic list could be more extensive. Please write more examples of testing the contact angle of rough surfaces. For example, the following papers address these issues:

https://doi.org/10.1038/s41598-020-80729-9

https://doi.org/10.1016/j.triboint.2021.107139

https://doi.org/10.1016/j.jcis.2004.02.036

Response 3: We add ” Tammar S. Meiron...axisymmetric heavy droplet.” in lines 36-42 to complete references.

Point 4: Line 53 - When we say the drop is heavy and when it's light? What values define it?

Response 4: The heavy droplet is the same as the large-volume droplet with the Bond number B0>0.1 . In lines 27 we simplify “an axisymmetric large-volume droplet on a microstructured surface in gravity” as “an axisymmetric heavy droplet”.

Point 5: Please write what parameters of the surface microstructure are important for the prediction of the contact angle? Examples of descriptions of surface microstructure parameters can be found, for example, in:

https://doi.org/10.3390/cryst11111371

https://doi.org/10.1007/s42242-019-00042-x

https://doi.org/10.1016/j.measurement.2022.112066

Response 5: We add ” The relative structure…our previous study ” in lines 209-211 to show the important parameters for the prediction of the contact angle.

Explain: Microstructure parameters are described as the periodic spacing a , the diameter d and the height h, which can be seen in figure 5.

Point 6: Chapter 4.1. -Provide the key processing parameters that affect the achievement of the appropriate surface microgeometry.

Response 6: Reply: We add ” Deep reactive iron…the passivation cycle” in lines 185-187 to show the key processing parameters.

Point 7: Specify the time of dispensing the drop and measuring the drop after deposition on the surface.

Response 7: We add” After emitting droplets for ten minute” in lines 197 to specify the time.

Reviewer 3 Report

This paper developed a numerical model to obtain the geometric parameters of a droplet on a micro-structured surface. The numerical model was established based on the arc differential method of the Bashforth-Adams equation and the principle of the total energy minimization. The apparent contact angles and the diameter of the solid-liquid interface of the droplets by the proposed model were in good agreement with those of experiments. However, some issues need to be clarified and addressed.

 1.      Fig. 4 and Fig. 8 are indistinct, the authors should redraw these figures.

2.      The explanations of some important symbols in equations were lacking. For example, in Eq. (9) and (10), is xv the radius of the solid-liquid interface? Is zv the height of the barycenter?

 3.      In Eq. (9) and (10), how the area of the solid-liquid interface of the droplet on a sphere-shaped surface? A detailed deduction result should be added.

 4.      In the calculation of the gravitational potential energy in Eq. (9) and (10), is the height of the barycenter (zv) a constant? Did the variation of zv of the droplets with different wetting morphologies be considered?

 5.      The authors used the numerical enumeration method to find the minimum total energy of the system, while did the Lagrange multiplier method with a constraint of the constant droplet volume was considered to solve the minimum energy? See Ref. (Colloids and Surfaces A 2020, 597, 124797; Langmuir 2003, 19, 4, 1249–1253)

Author Response

Dear Reviewer

Thank you for raising these issues and we have revised the document for your review in the light of your comments. I look forward to hearing from you.

Best regards

    Response to Reviewer 3 Comments

Point 1:  Fig. 4 and Fig. 8 are indistinct, the authors should redraw these figures.

Response 2: Fig. 4 and Fig. 8 have been rewritten.

Point 2: The explanations of some important symbols in equations were lacking. For example, in Eq. (9) and (10), is xv the radius of the solid-liquid interface? Is zv the height of the barycenter?

Response 2: Yes , zv  and xv  are the height and the contact circle radius of the droplet, respectively, solved at the volume constraint. We add ”  xv and zv  ... at the volume constraint” in lines 159-160 to show it.

Point 3:  In Eq. (9) and (10), how the area of the solid-liquid interface of the droplet on a sphere-shaped surface? A detailed deduction result should be added.

Response 3: We add Eq. (14) and Eq. (15) in lines 157-158, in which the areas of the solid-liquid interface of the droplet on a sphere-shaped surface are showed.

Point 4: In the calculation of the gravitational potential energy in Eq. (9) and (10), is the height of the barycenter (zv) a constant? Did the variation of zv of the droplets with different wetting morphologies be considered?

Response 4: No ,  zv  is the height of the droplet and zv  is a the variant determined by different b(the size factor) to conform to the droplet volume. The wetting morphologies only affect the total energy of the droplet. The wetting morphologies do not affect zv .

Point 5: The authors used the numerical enumeration method to find the minimum total energy of the system, while did the Lagrange multiplier method with a constraint of the constant droplet volume was considered to solve the minimum energy? See Ref. (Colloids and Surfaces A 2020, 597, 124797; Langmuir 2003, 19, 4, 1249–1253)

Response 5: We add ”The curvature radii...the minimum energy” in lines 133-137 to explain this question.  

Round 2

Reviewer 2 Report

All major corrections have been made by the Authors. The publication is acceptable.